# Learned Neural Network Representations are Spread Diffusely with Redundancy

## Abstract

Representations learned by pre-training a neural network on a large dataset are increasingly used successfully to perform a variety of downstream tasks. In this work, we take a closer look at how features are encoded in such pre-trained representations. We find that learned representations in a given layer exhibit a degree of *diffuse redundancy*, *i.e.*, any randomly chosen subset of neurons in the layer that is larger than a threshold size shares a large degree of similarity with the full layer and is able to perform similarly as the whole layer on a variety of downstream tasks. For example, a linear probe trained on 20% of randomly picked neurons from a ResNet50 pre-trained on ImageNet1k achieves an accuracy within 5% of a linear probe trained on the full layer of neurons for downstream CIFAR10 classification. We conduct experiments on different neural architectures (including CNNs and Transformers) pre-trained on both ImageNet1k and ImageNet21k and evaluate a variety of downstream tasks taken from the VTAB benchmark. We find that the loss & dataset used during pre-training largely govern the degree of diffuse redundancy and the "critical mass" of neurons needed often depends on the downstream task, suggesting that there is a task-inherent sparsity-performance Pareto frontier. Our findings shed light on the nature of representations learned by pre-trained deep neural networks and suggest that entire layers might not be necessary to perform many downstream tasks. We investigate the potential for exploiting this redundancy to achieve efficient generalization for downstream tasks and also draw caution to certain possible unintended consequences.

## 1 Introduction

Over the years, many architectures such as VGG (Simonyan & Zisserman, 2014), ResNet (He et al., 2016), and Vision Transformers (ViTs) (Kolesnikov et al., 2021) have been proposed that achieve competitive accuracies on many benchmarks including the ImageNet (Russakovsky et al., 2015) challenge. A key reason for the success of these models is their ability to learn useful representations of data (LeCun et al., 2015).

Prior works have attempted to understand representations learned by deep neural networks through the lens of mutual information between the representations, inputs and outputs (Shwartz-Ziv & Tishby, 2017) and hypothesize that neural networks perform well because of a "compression" phase where mutual information between inputs and representations decreases. Moreover recent works on interpretability have found that many neurons in learned representations are *polysemantic*, *i.e.*, one neuron can encode multiple "concepts" (Elhage et al., 2022; Olah et al., 2020), and that one can then train sparse linear models on such concepts to do "explainable" classification (Wong et al., 2021). However, it is not well understood if or how extracted features are concentrated or spread across the full representation.

While the length of the feature vectors extracted from state-of-the-art networks [1] can vary greatly, their accuracies on downstream tasks are not correlated to the size of the representation (see Table 1), but rather depend mostly on the inductive biases and training recipes (Wightman et al., 2021; Steiner et al., 2021). In all cases, the size of extracted feature vector (*i.e.* number of neurons) is orders of

---

[1]Extracted features for the purpose of this paper refers to the representation recorded on the penultimate layer, but the larger concept applies to any layer

Table 1: Different model architectures with varying penultimate layer lengths trained on ImageNet1k. WRN50-2 stands for WideResNet50-2. Implementation of architectures is taken from `timm` (Wightman, 2019). Diffused redundancy here measures what fractions of neurons (randomly picked) can be discarded to achieve within $\delta = 90\%$ performance of the full layer.

| Model | Feature Length | ImageNet1k Top-1 Accuracy | Diffused Redundancy for $\delta = 0.9$ | | | |
|---|---|---|---|---|---|---|
| | | | CIFAR10 | CIFAR100 | Flowers | Oxford-IIIT-Pets |
| ViT S-16 | 384 | 64.82% | 0.70 | 0.50 | 0.50 | 0.80 |
| ViT S-32 | 384 | 55.73% | 0.70 | 0.50 | 0.50 | 0.70 |
| ResNet18 | 512 | 69.23% | 0.80 | 0.50 | 0.50 | 0.90 |
| ResNet50 | 2048 | 80.07% | 0.90 | 0.50 | 0.20 | 0.90 |
| WRN50-2 | 2048 | 77.00% | 0.95 | 0.80 | 0.50 | 0.95 |
| VGG16 | 4096 | 73.36% | 0.95 | 0.80 | 0.80 | 0.95 |

magnitude less than the dimensionality of the input (*e.g.*ImageNet models, the inputs are $224 \times 224 \times 3 = 150528$ dimensional) and thus allows efficient transfer to many downstream tasks (Kolesnikov et al., 2020; Bengio et al., 2013; Pan & Yang, 2009; Tan et al., 2018). We show that even when using a *random* subset of these extracted neurons one can achieve downstream transfer accuracy close to that achieved by the full layer, thus showing that learned representations exhibit a degree of redundancy (Table 1).

Early works in perception suggest that there are many redundant neurons in the human visual cortex (Attneave, 1954) and some works argued that a similar redundancy in artificial neural networks should help in faster convergence (Izui & Pentland, 1990). In this paper we revisit redundancy in the context of modern DNN architectures that have been trained on large-scale datasets. In particular, we propose the **diffused redundancy** hypothesis and systematically measure its prevalence across different pre-training datasets, losses, model architectures and downstream tasks. We also show how this kind of redundancy can be exploited to obtain desirable properties such as generalization performance and better parity in inter-class performance. We highlight the following contributions:

- We present the diffused redundancy hypothesis which states that learned representations exhibit redundancy that is diffused throughout the layer. Our work aims to better understand the nature of representations learned by DNNs.

- We propose a measure of diffused redundancy and systematically test our hypothesis across various architectures, pre-training datasets & losses and downstream tasks.
  - We find that diffused redundancy is significantly impacted by pre-training datasets & loss and downstream datasets.
  - We find that models that are explicitly trained such that particular parts of the full representation perform as well as the full layer, *i.e.*, these models have *structured redundancy* (*e.g.* (Kusupati et al., 2022)), also exhibit a significant amount of diffused redundancy, showing that this phenomenon is perhaps inevitable when DNNs have a wide enough final layer.
  - We quantify the degree of diffused redundancy as a function of the number of neurons in a given layer. As we reduce the dimension of the extracted feature vector and re-train the model, the degree of diffused redundancy decreases significantly, implying that diffused redundancy only appears when the layer is wide enough to accommodate redundancy.

- Finally we draw caution to some potential undesirable side-effects of exploiting diffused redundancy for efficient transfer learning that have implications for fairness.

## 1.1 RELATED WORK

Closest to our work is that of Dalvi et al. (2020) who also investigate neuron redundancy but in the context of pre-trained language models. They analyze two language models and find that they can achieve good downstream performance with a significantly smaller subset of neurons. However, there are two key differences to our work. First, their analysis of neuron redundancy uses neurons from all layers (by concatenating each layer), whereas we show that such redundancy exists even at the level of a single (penultimate) layer. Second, and perhaps more importantly, they use feature selection to choose the subset of neurons, whereas we show that features are diffused throughout and that even a *random* pick of neurons suffices. Our work also differs by analyzing vision models

(instead of language models) and using a diverse set of 21 pre-trained models (as opposed to testing only two models) which allows us to better understand the causes of such redundancy.

**Efficient Representation Learning** These works aim to learn representations which are "slim", with the goal of efficient deployment on edge devices (Yu et al., 2018; Yu & Huang, 2019; Cai et al., 2019). Recently proposed paradigm of *Matryoshka Representation Learning* (Kusupati et al., 2022) aims to learn nested representations where one can perform downstream tasks with only a small portion of the representation. The goal of such representations is to allow quick, adaptive deployment without having to perform multiple, often expensive, forward passes. These works could be seen as inducing *structured redundancy* on the learned representations, where pre-specified parts of the representation are made to perform similar to the full representation. Our work, instead, aims to looks at *diffused redundancy* that arises naturally in the training of DNNs. We carefully highlight the tradeoffs involved in exploiting this redundancy.

**Pruning and Compression** Many prior works focus on pruning weights (LeCun et al., 1989; Han et al., 2015; Frankle & Carbin, 2019; Hassibi & Stork, 1992; Levin et al., 1993; Dong et al., 2017; Lee et al., 2018) and how it can lead to sparse neural networks with many weights turned off. Our focus, however, is on understanding redundancy at the neuron level, without changing the weights. Work on structured pruning is more closely related to our work Li et al. (2016); He et al. (2017), however, a key focus of these works is to prune channels/filters from convolution layers. Our work is more focused on understanding the nature of learned features and is more broadly applicable to all kinds of layers and models. We additionally focus on randomly pruning neurons, whereas structured pruning methods perform magnitude or feature-selection based pruning.

**Explainability/Interpretability** Many works aim to understand learned representations with the goal of better explainability (Mahendran & Vedaldi, 2014; Yosinski et al., 2015; Alain & Bengio, 2018; Kim et al., 2018; Olah et al., 2017; 2020; Elhage et al., 2022; Zeiler & Fergus, 2013). However, two works are especially related to our work: sparse linear layers (Wong et al., 2021) which show that one can train sparse linear layers on top of extracted features from DNNs; and concept bottleneck models (Koh et al., 2020) which explicitly introduce a layer in which each neuron corresponds to a meaningful semantic concept. Both these works explicitly optimize for small/sparse layers, whereas our work shows that similar "small" layers already exist in pre-trained networks, and in fact, can be found simply with random sampling.

**Understanding Deep Learning** A related concept is that of instrinsic dimensionality of DNN landscapes (Li et al., 2018). Similar to our work, intrinsic dimensionality also requires dropping random parameters (weights) of the network. We, however, are concerned with dropping individual neurons. Other works on understanding deep learning (Shwartz-Ziv & Tishby, 2017; Achille & Soatto, 2017) have also looked at the learned features, however none of these works analyze the redundancy at neuron level.

## 2 THE DIFFUSED REDUNDANCY PHENOMENON

Prior observations about a *compression* phase (Shwartz-Ziv & Tishby, 2017) suggest that the representations need not store a lot of information about the input; and observations about there being *polysemantic* neurons (Olah et al., 2020) states that one neuron can store multiple *concepts*. Both findings allude to the possibility of not needing all neurons in the learned feature space. Extending these observations, we propose the *diffused redundancy* hypothesis:

*Learned features are diffused throughout a given layer with redundancy such that a randomly chosen subset of neurons can perform similar to the whole layer for a variety of downstream tasks.*

Note that our hypothesis has two related but distinct parts to it: 1) redundancy in learned features, and 2) diffusion of this redundancy throughout the extracted feature vector. *Redundancy* refers to features being replicated in parts of the representation so that one can perform downstream tasks with parts of representation as well as with the full representation. *Diffusion* refers to this redundancy being spread all over the feature vector (as opposed to being structured), *i.e.*, *any* random subset (of sufficient size) of the feature vector performs equally well.

In order to evaluate the *redundancy* part of the diffused redundancy hypothesis we use two tasks: 1) representation similarity between randomly chosen subsets of a representation with the whole

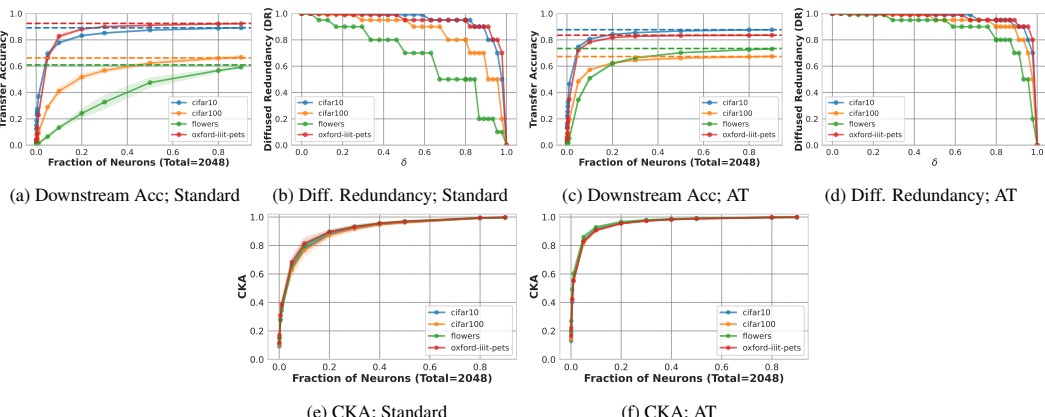

Figure 1: **[Testing For Diffused Redundancy in ResNet50 Pre-trained on ImageNet1k]** Top: transfer accuracies + DR measure (Eq 1) on different downstream datasets, dotted horizontal line shows accuracy obtained using the full layer. We see that accuracy obtained using parts of representation varies greatly with pre-training loss (much more diffused redundancy in adversarially trained ResNet), but also depends on the downstream dataset. Bottom: comparing CKA between a randomly chosen fraction of neurons to the whole layer. Here we evaluate CKA on samples from different datasets and find that similarity of a subset of layer rapidly increases, reaching a similarity of greater than 90% on the adversarially trained ResNet50 with only 10% randomly chosen neurons. All values are averaged over 5 random picks and errors bars show standard deviation.

representation, and 2) transfer accuracy on out-of-distribution datasets (using a linear probe) of randomly chosen subsets of the representation compared to the whole representation. To estimate *diffusion*, we run each check for redundancy over multiple random seeds and plot the standard deviation over these runs.

**Representation Similarity of Part vs Whole** Centered Kernel Alignment (CKA) is a widely used representation similarity measure and takes in two representations of $n$ data points $Z \in \mathbb{R}^{n \times d_1}$ and $Y \in \mathbb{R}^{n \times d_2}$ and gives a similarity score between 0 and 1 (Kornblith et al., 2019). Intuitively, CKA (with linear kernel, see Appendix A for details about CKA) measures if the two representations rank the $n$ points similarly (where similarity is based on cosine distances). For a given neural network $g$ and $n$ samples drawn from a given data distribution, *i.e.*, $X \sim \mathcal{D}$, let $g(X)$ be the (penultimate) layer representation. If $m$ is a boolean vector representing a subset of neurons in $g(X)$, then we aim to measure CKA($m \odot g(X), g(X)$) to estimate how much redundancy exists in the layer. If indeed CKA($m \odot g(X), g(X)$) is high (*i.e.* close to 1) then it's a strong indication that the diffused redundancy hypothesis holds.

**Downstream Transfer Performance of Part vs Whole** A commonly used paradigm to measure the quality of learned representations is to measure their performance on a variety of downstream tasks (Zhai et al., 2019; Kolesnikov et al., 2020). Here, we attach a linear layer ($h$) on top of the extracted features of a network ($g$) to do classification. This layer is then trained using the training dataset of the particular task (keeping $g$ frozen). If features were to be diffused redundantly then accuracy obtained using $h \circ g$, *i.e.* linear layer attached to the entire feature vector, should be roughly the same as $h' \circ (m \odot g)$; where $m$ is a boolean vector representing a subset of neurons extracted by $g$, and $h$ & $h'$ are independently trained linear probes.

For both tasks, *i.e.* representation similarity and downstream transfer performance, we evaluate on CIFAR10/100 (Krizhevsky et al., 2009), Oxford-IIIT-Pets (Parkhi et al., 2012) and Flowers (Nilsback & Zisserman, 2008) datasets, from the VTAB benchmark (Zhai et al., 2019). Training and pre-processing details are included in Appendix B.

**Measure of Diffused Redundancy** In order to rigorously test our hypothesis, we define a measure of diffused redundancy ($DR$) for a given model ($g$) with $\mathcal{M}$ being a set of all possible boolean vectors of size $|g|$, *i.e.* size of the representation extracted from $g$. Each vector $m \in \mathcal{M}$ represents a possible subset of neurons from the entire layer. This measure is defined on a particular task ($T$) as follows:

$$DR(g, T, \delta) = 1 - \frac{\min f \text{ , s.t. } \frac{1}{|\mathcal{M}_f|} \sum_{m \in \mathcal{M}_f} \frac{T(m \odot g)}{T(g)} \geq \delta}{|g|}, \qquad (1)$$

$$\mathcal{M}_f = \left\{ m \in \mathcal{M} | \sum_i m_i = f \; ; \; m \in \{0, 1\}^{|g|} \right\}$$

Here $T(.)$ denotes the performance of the model inside () for the particular task and $\delta$ is a user-defined tolerance level. For the task for representation similarity $T(m \odot g)$ is CKA between subset of neurons denoted by $m \odot g$ and $g$, and $T(g)$ is always 1, since it denotes CKA between $g$ and $g$. For downstream transfer performance, $T(m \odot g)$ is the test accuracy obtained by training a linear probe on the portion of representation denoted by $m \odot g$ and $T(g)$ is the test accuracy obtained using the full representation. For $\delta = 1$, this measure tells what fraction of neurons could be discarded to exactly match the performance of the entire set of neurons. A higher value of $DR$ denotes that fewer random neurons could match the task performance of the full set of neurons, and thus indicates higher redundancy. Since $\mathcal{M}$ contains an exponential number of vectors ($2^{|g|}$), precisely estimating this quantity is hard. Thus, we first choose a few $f$ (number of neurons to be chosen) to define subsets of $\mathcal{M}$. Then for each $\mathcal{M}_f$ we randomly select 5 samples.

## 2.1 Prevalence of Diffused Redundancy in Pre-Trained Models

Figure 1 checks for diffused redundancy in the penultimate layer representation of two types of ResNet50 pre-trained on ImageNet1k: one using the standard cross-entropy loss and another trained using adversarial training (Madry et al., 2019) (with $\ell_2$ threat model and $\epsilon = 3$). We check for diffused redundancy using both tasks of representation similarity and downstream transfer performance.

**Redundancy** This is indicated along the x-axis of Fig 1, *i.e.*, redundancy is shown when some small subset of the full set of neurons can achieve almost as good performance as the full set of neurons. When looking at downstream task performance (Figs 1a&1c), in order to obtain performance within some $\delta\%$ of the full layer accuracy (dotted lines), the fraction of neurons that can be discarded are task-dependent, *e.g.* across both training types we see that flowers (102 classes) and CIFAR100 (100 classes) require more fraction of neurons than CIFAR10 (10 classes) and oxford-iiit-pets (37 classes), perhaps because both these tasks have more classes. Additionally, across all datasets, the model trained with adversarial training exhibits more diffused redundancy (Fig 1b&1d) than the one trained with standard loss, meaning we can discard far more neurons for the adversarially trained model to reach close to the full layer accuracy. Interestingly when looking at CKA between part of the feature vector with the full extracted vector (Figs 1e&1f), we do not see a significant difference in trends when evaluating CKA on samples from different datasets. However, we still see that we can achieve a given level of CKA with far fewer fraction of neurons in the adversarially trained ResNet50 as compared to the usually trained ResNet50.

**Diffusion** This is indicated by small error bars in Figs 1a&1c&1e&1f. If redundancy were instead very structured, then different random picks of neurons would have high variance, however the error bars here are very low, showing that performance is very stable across different random picks, thus indicating that redundancy is diffused throughout the layer.

While both tasks of downstream transfer and CKA between part and whole indicate higher diffused redundancy for the adversarially trained model, we see that downstream transfer performance can differ substantially based on the dataset (while CKA remains fairly stable across the same datasets), indicating that downstream performance turns out to be a "harder" test for diffused redundancy. Thus, in the rest of the paper we examine diffused redundancy through the lens of downstream transfer performance and include CKA results in Appendix A.

## 2.2 Understanding Why *Any* Random Subset Works

Many prior works explicitly train models to that have "small" representations (*e.g.* (Kusupati et al., 2022; Yu et al., 2018; Yu & Huang, 2019; Cai et al., 2019) with the goal of efficient downstream learning. These works show that, when explicitly optimized, networks can perform downstream classification with fewer neurons than typically used in state-of-the-art architectures. We show, however, that such subsets already exist in models that are not explicitly trained for this goal and

in fact one doesn't even have to try hard to find this subset; it can be *randomly* chosen. Later in section 3.3 we compare some of these efficient representation learning methods to randomly chosen subsets and carefully analyze the tradeoffs involved. Here, however, we seek to better understand why (almost) *any* random subset works.

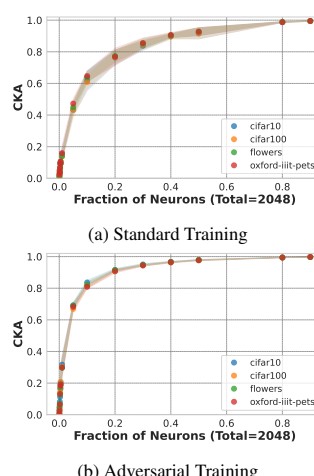

(a) Standard Training

(b) Adversarial Training

Figure 2: **[Why Any Random Subset Works]** Similarity between any two randomly picked sets of $k\%$ neurons becomes fairly high (for a "critical mass" of $k\%$), thus showing that any random pick beyond this threshold is likely to perform similarly.

We analyze this through the lens of representation similarity (Kornblith et al., 2019). More specifically, we calculate CKA between two random picks of $k\%$ neurons in the penultimate layer (averaged over 10 such randomly picked pairs) on samples taken from different datasets. Fig 2 shows CKA results averaged over these different picks of pairs of subsets of the full set of neurons. We see that after a picking a certain threshold, *i.e.* for a large enough value of $k$, the similarity between any two randomly picked pairs of heads is fairly high. For example, for the adversarially trained ResNet50 (Fig 2b), we observe that any $10\%$ of neurons picked from the penultimate layer are highly similar (CKA of about $0.8$), with very low error bars. A similar value of CKA is obtained with $20\%$ of neurons for the standard ResNet50 model. These results indicate that, given a sufficient size, picking any random subset of that size has very similar representations and this provides some intuition for why *any* random subset works equally well.

## 3 FACTORS INFLUENCING THE DEGREE OF DIFFUSED REDUNDANCY

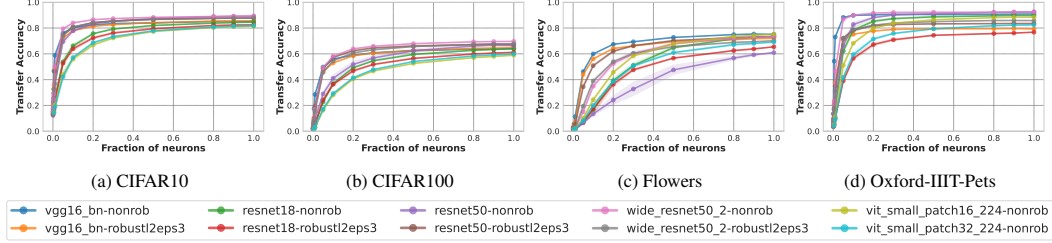

(a) CIFAR10          (b) CIFAR100          (c) Flowers          (d) Oxford-IIIT-Pets

Figure 3: **[Comparisons Across Architectures For Downstream Task Accuracy]** All models shown here are pre-trained on ImageNet1k. We see that diffused redundancy exists across architectures, and the trend observed in Figure 1c&1a regarding adversarially trained models also holds here as models curves that are more "inside" are the ones trained with standard loss.

In order to better understand the phenomenon of diffused redundancy we analyze 21 different pre-trained models, with different architectures, pre-training datasets and losses. We then evaluate each model for transfer accuracy on 4 datasets mentioned in Section 2.

**Architectures** We consider VGG16 (Simonyan & Zisserman, 2014), ResNet18, ResNet50, WideResNet50-2 (He et al., 2016), ViT-S16 & ViT-S32 (Kolesnikov et al., 2021). Additionally we consider ResNet50 with varying widths of the final layer (denoted by `ResNet50_ffx` where x denotes the number of neurons in the final layer).

**Upstream Datasets** ImageNet-1k & ImageNet-21k Russakovsky et al. (2015).

**Upstream Losses** Standard cross-entropy, adversarial training ($\ell_2$ threat model, $\epsilon = 3$) (Madry et al., 2019), MRL Loss (Kusupati et al., 2022).

**Downstream Datasets** CIFAR10/1000, Oxford-IIIT-Pets, and Flowers, same as Section 2.

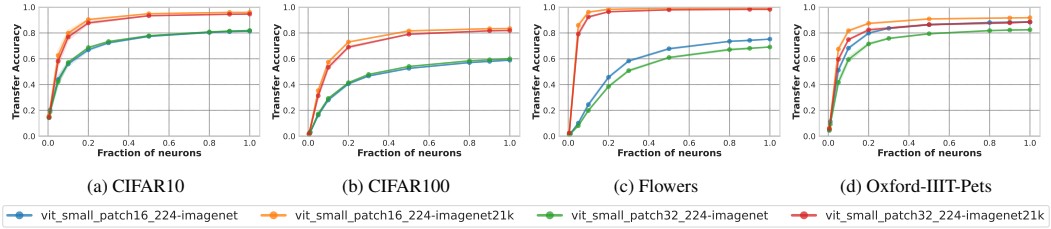

(a) CIFAR10     (b) CIFAR100     (c) Flowers     (d) Oxford-IIIT-Pets

Figure 4: **[Comparison Across Upstream Datasets]** We see that degree of diffused redundancy depends a great deal on the upstream training dataset, in particular models trained on ImageNet21k exhibit a higher degree of diffused redundanacy, although the differences in the degree of diffused redundanacy are downstream task dependent

Pre-trained weights for adversarially trained models were taken from (Salman et al., 2020). Weights for ViTs (both ImageNet21k and ImageNet1k) were taken from the code released by (Kolesnikov et al., 2021; Steiner et al., 2021). All `ResNet50_ffx` and the ResNet50 trained with MRL loss were taken from code released by (Kusupati et al., 2022), and all standard models were taken from `timm` and `torchvision` (Wightman, 2019; Paszke et al., 2019).

## 3.1 Effects of Architecture, Upstream Loss, Upstream Datasets, and Downstream Datasets

Extending the analysis in Section 2, we evaluate the diffused redundancy hypothesis on other architectures. Fig 3 shows transfer performance for different architectures. All architectures shown in Fig 3 are trained on ImageNet1k. We find that our takeaways from Section 2 also extend to other architectures.

Fig 4 compares two instances each of ViT-S16 and ViT-S32, one trained on a bigger upstream dataset (ImageNet21k) and another on a smaller dataset (ImageNet1k)

Note that nature of all curves in both Figs 3&4 highly depends on downstream datasets. This is also consistent with the initial observation of Section 2 about diffused redundancy being downstream dataset dependent.

## 3.2 Diffused Redundancy as a Function of Layer Width

We take the usual ResNet50 with a penultimate layer consisting of 2048 neurons and compare it with variants that are pre-trained with a much smaller penultimate layer, these are denoted by `ResNet50_ffx` where x ($< 2048$) is the number of neurons in the penultimate layer. Fig 5 shows how diffused redundancy slows fades away as we squeeze the layer to be smaller. In fact, for `ResNet50_ff8`, we see that across all datasets we need $> 90\%$ of the full layer to achieve performance close to the full layer. This shows that diffused redundancy only appears in DNNs when the layer is sufficiently wide to encode redundancy.

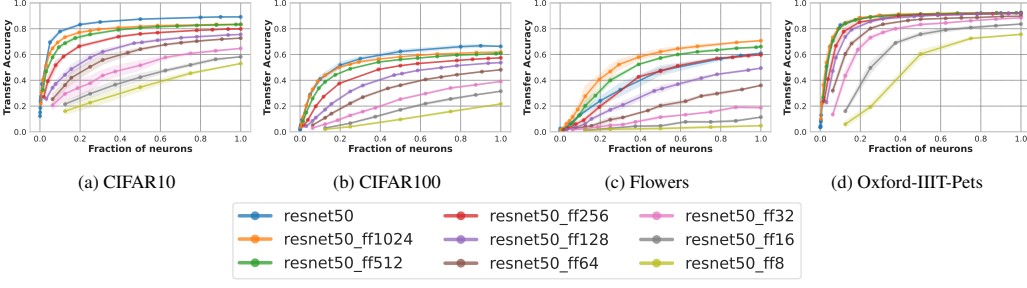

(a) CIFAR10     (b) CIFAR100     (c) Flowers     (d) Oxford-IIIT-Pets

Figure 5: **[Diffused Redundancy as Function of Layer Width]** As we make the length of layer smaller, the degree of redundancy becomes lesser. For `ResNet50_ff8`, *i.e.*ResNet50 with only 8 neurons in final layer, we see that across all datasets we need almost $90\%$ of neurons to achieve similar accuracy as the full layer.

### 3.3 COMPARISON WITH METHODS THAT OPTIMIZE FOR LESSER NEURONS

Matryoshka Representation Learning (MRL) is a recently proposed paradigm which learns nested representations such that first $k, 2k, 4k, ..., N$ (where $N$ = size of the full layer) dimensions of the extracted feature vector are all explicitly made to be good at minimizing upstream loss, with the intuition of learning coarse-to-fine representations. This ensures that one can flexibly use these smaller parts of the representation for downstream tasks. MRL, thus, ensures that redundancy shows up in learned representations in a *structured* way, *i.e.*, we know the first $k, 2k, ...$ neurons can be picked and used for downstream tasks and should perform reasonably.

Here we investigate two questions regarding Matryoshka representations: 1) do these representations also exhibit the phenomenon of diffused redundancy? *i.e.* if we were to ignore the structure imposed by MRL-type training and instead just pick random neurons from all over the layer, do we still get reasonable performance?, and 2) how do they compare to representations learned by other kinds of losses?

Figure 6 investigates these questions by comparing ResNet50 representations learned using MRL loss to other losses. `resnet50_mrl_nonrob_first` (red line) denotes resnet50 learned using MRL loss and evaluated on parts of representation that were optimized to have low upstream loss (*i.e.* first $k, 2k, ...N$ neurons, here $k = 8$ and $N = 2048$) and `resnet50_mrl_nonrob_random` (green line) refers to the same model with same number of neurons chosen for evaulation, except they're chosen at random from the entire layer.

First, we interestingly see that even the ResNet50 trained with MRL loss exhibits diffused redundancy (denoted by green line spiking very quickly for most datasets in Fig 6), despite having been trained to only have structured redundancy. Based on this observation we conjecture that diffused redundancy is a natural consequence of having a wide layer. Second, we see that ResNet50 trained on MRL indeed does better in the low neuron regime across datasets (red line on the extreme left part of the plots in Fig 6), but other models quickly catch up as we pick more neurons, thus indicating that major efficiency benefits of MRL-type models are best realized when using extremely low number of neurons, else one can obtain similar downstream performances by simply picking random samples from existing pre-trained models.

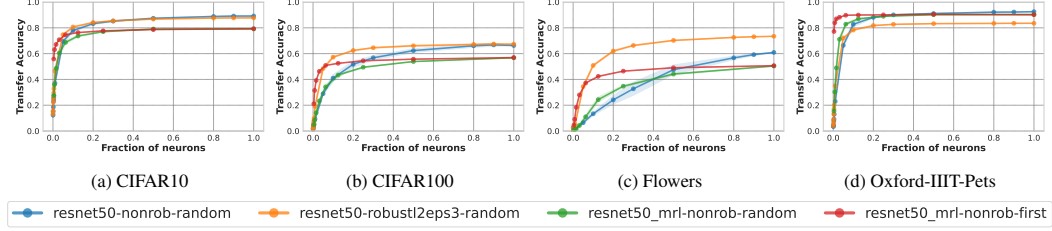

| (a) CIFAR10 | (b) CIFAR100 | (c) Flowers | (d) Oxford-IIIT-Pets |
| --- | --- | --- | --- |

Figure 6: **[Comparison of Diffused Redundancy in MRL vs other losses]** Here we compare ResNet50 trained using multiple losses including MRL (Kusupati et al., 2022). Red line shows results for part of the representation explicitly optimized in MRL, whereas green line shows results for parts that are picked randomly from the same representation. Even the MRL model shows a significant amount of diffused redundancy despite being explicitly trained to instead have structured redundancy.

## 4 POSSIBLE FAIRNESS-EFFICIENCY TRADEOFFS IN EFFICIENT DOWNSTREAM TRANSFER

One natural use-case for diffused redundancy is efficient transfer to downstream datasets, *i.e.*, use only a random subset instead of the entire feature vector. This would lead to faster training and lesser storage requirements since it reduces number of additional parameters. As defined in Eq 1 and as also seen in prior works (*e.g.* Kusupati et al. (2022)) dropping neurons comes at a small cost in performance as compared to the full set of neurons. Here we take a deeper look into this drop in overall performance and investigate how it is distributed across classes. If the drop affects only a few classes, then dropping neurons – although efficient for downstream tasks – could have implications for fairness, which is not only of concern to ML researchers and practitioners (Zafar

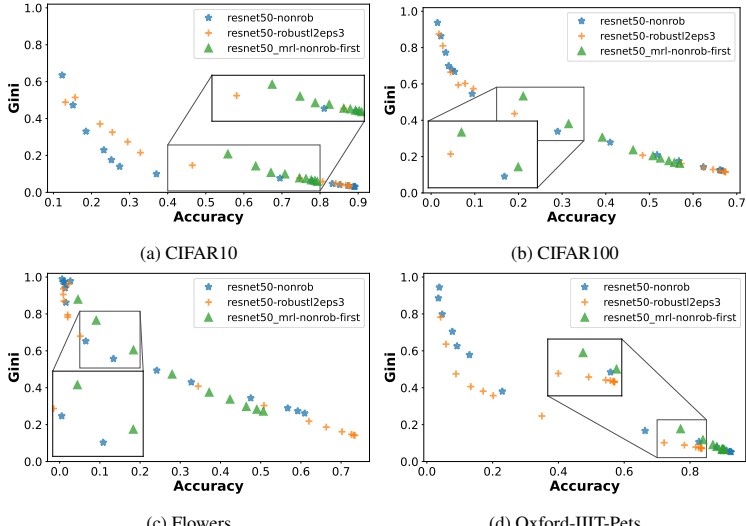

Figure 7: **[Gini Coefficient of Class-Wise Accuracies as we Drop Neurons]** Higher value of Gini coefficient indicates higher inequality (Gini, 1921). We see that for all models gini coefficients become higher as the accuracy reduces (as a result of dropping neurons). Additionally in some regions (highlighted in the plots), the model explicitly optimized for efficient transfer (`resnet50_mrl`) can give rise to higher gini values, resulting in a more unequal spread of accuracy over classes.

et al., 2017; Hardt et al., 2016; Holstein et al., 2019), but also to lawyers Tolan et al. (2019) and policymakers (Veale et al., 2018).

We compare the spread of accuracies across classes using inequality indices, which are commonly used in economics to study income inequality (De Maio, 2007; Schutz, 1951) and have also recently been adopted in the fair ML literature (Speicher et al., 2018). We use gini index (Gini, 1921) and coefficient of variation (Lawrence, 1997) to quantify the spread of performance across classes. For a perfect spread, both gini and coefficient of variation are 0, and higher values indicate higher inequality.

Figure 7 compares the gini index for various models at varying levels of accuracy (note that accuracy monotonically increases with more neurons, hence right more point for a model represents the model with all neurons). We make two observations: across all datasets and all models we find that a loss in accuracy (compared to the full layer) comes at the cost of a few classes, as opposed being smeared throughout classes, as indicated by high gini values on the left of each plot. Additionally, we observe that the model trained using MRL loss tends to have slightly higher gini values in the regions where drop in accuracy is slightly higher (highlighted on the plots). Similar trends are also observed with coeff. of variation as shown in Appendix C. These results draw caution to potential unintended side-effects of exploiting diffused redundancy and suggest that there could be a possible fairness-efficiency tradeoff involved.

## 5 CONCLUSION AND BROADER IMPACTS

We introduce the diffused redundancy hypothesis and analyze a wide range of models with different upstream training datasets & losses, architectures. We carefully analyze the causes of such redundancy and find that upstream training (both loss and datasets) plays a crucial role and that this redundancy also depends on the downstream dataset. One direct practical consequence of our observation is increased efficiency for downstream training times which can have many positive impacts in terms of reduced energy costs (Strubell et al., 2019) which is crucial in moving towards "green" AI (Schwartz et al., 2020). We, however, also draw caution to potential pitfalls of such efficiency gains in Section 4, which might hurt the accuracy of certain classes more than others, thus having direct consequences for fairness. We see our work as a contribution to the scholarship on better understanding deep learning through an empirical lens, while also highlighting possible pitfalls.

REPRODUCIBILITY STATEMENT

We include all training and pre-processing details in Appendix B. We have also attached all the code used to generate the results in our paper. All of our evaulation is based on standard publicly available datasets which we have cited throughout the paper (and our code contains scripts to automatically download these datasets). Appendix B also includes links to all pre-trained models (with proper citations) so that exact numbers can be replicated.

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

## A    MEASURING DIFFUSED REDUNDANCY

### A.1    CKA DEFINITION

In all our evaluations we use CKA with a linear kernel (Kornblith et al., 2019) which essentially amounts to the following steps:

1. Take two representations $Y \in \mathbb{R}^{n \times d1}$ and $Z \in \mathbb{R}^{n \times d2}$
2. Compute dot product similarity within these representation, *i.e.* compute $K = YY^T$, $L = ZZ^T$
3. Normalize $K$ and $L$ to get $K' = HKH$, $L' = HLH$ where $H = I_n - \frac{1}{n}\mathbf{11}^T$
4. Return $\text{CKA}(Y, Z) = \frac{\text{HSIC}(K,L)}{\sqrt{\text{HSIC}(K,K)\text{HSIC}(L,L)}}$, where $\text{HSIC}(K, L) = \frac{1}{(n-1)^2}\big(\text{flatten}(K') \cdot \text{flatten}(L')\big)$

We use the publicly available implementation of Nanda et al. (2022), which provides an implementation that can be calcuated over multiple mini-batches: `https://github.com/nvedant07/STIR`

### A.2    ADDITIONAL CKA RESULTS

Fig 8 shows CKA comparison between randomly chosen parts of the layer and the full layer for different kinds of ResNet50. We observe that even ResNet50 trained with MRL loss shows a significant amount of diffused redundancy.

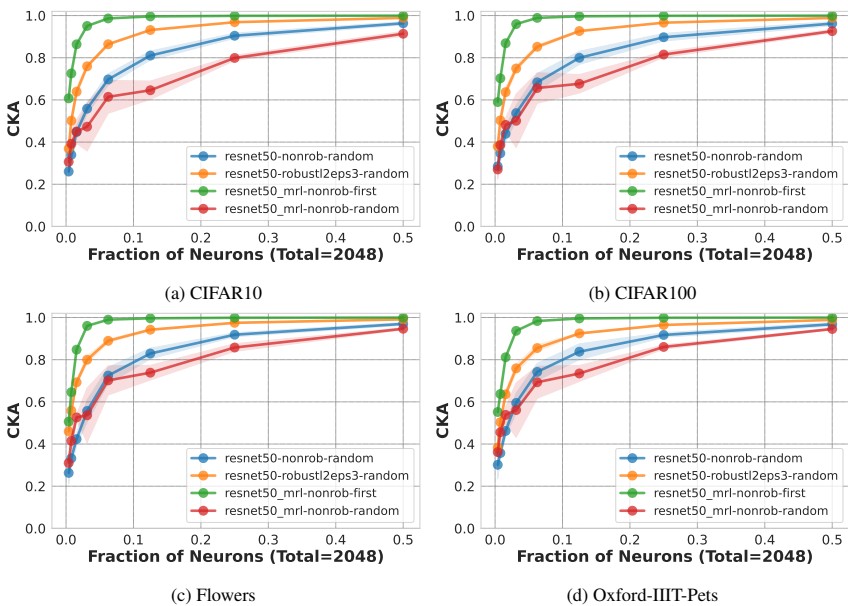

Figure 8: **[Comparison of Diffused Redundancy in MRL vs other losses, through the lens of CKA]** We see a similar trend as reported in Fig 6 in the main paper, where even the MRL model shows a significant amount of diffused redundancy despite being explicitly trained to instead have structured redundancy. The amount of diffused redundancy however is much lesser than the resnets trained using the standard loss and adv. training as denoted by a much lower red line across all datasets.

## B    TRAINING AND PRE-PROCESSING DETAILS FOR REPRODUCIBILITY

Here we list the sources of weights for the various pre-trained models used in our experiments:

- ResNet18 trained on ImageNet1k using standard loss: taken from `timm` v0.6.1.

- ResNet18 trained on ImageNet1k with adv training: taken from Salman et al. (2020):
- ResNet50 trained on ImageNet1k using standard loss: taken from `timm` v0.6.1.
- ResNet50 trained on ImageNet1k with adv training: taken from Salman et al. (2020): https://github.com/microsoft/robust-models-transfer.
- ResNet50 trained on ImageNet1k using MRL and with different final layer widths (`resnet50_ffx`): taken from released weights of by Kusupati et al. (2022): https://github.com/RAIVNLab/MRL.
- WideResNet50-2 on ImageNet1k both standard and avd. training: taken from Salman et al. (2020): https://github.com/microsoft/robust-models-transfer.
- VGG16 trained on ImageNet1k with standard loss: taken from `timm` v0.6.1.
- VGG16 trained on ImageNet1k with adv training: taken from  Salman et al. (2020): https://github.com/microsoft/robust-models-transfer.
- ViTS32 & ViTS16 trained on ImageNet21k & ImageNet1k: taken from weights released by Steiner et al. (2021): https://github.com/google-research/vision_transformer.

All linear probes trained on the representations of these models are trained using SGD with a learning rate of $0.1$, momentum of $0.9$, batch size of $256$, weight decay of $1e - 4$. The probe is trained for $50$ epochs with a learning rate scheduler that decays the learning rate by $0.1$ every $10$ epochs. Scripts for training can also be found in the attached code.

For pre-processing, we re-size all inputs to 224x224 (size used for pre-training) and apply the usual composition of RandomHorizontalFlip, ColorJitter(brightness=0.25, contrast=0.25, saturation=0.25, hue=0.25), RandomRotation(degrees=2). All inputs were mean normalized. For imagenet1k pre-trained models: mean = [0.485, 0.456, 0.406] and std = [0.229, 0.224, 0.225]. For imagenet21k pre-trained models: mean = [0.5,0.5,0.5], std = [0.5,0.5,0.5].

## C    COEFF OF VARIATION FOR MEASURING INEQUALITY IN INTER-CLASS ACCURACY

Fig 9 shows results for the same analysis shown in Fig 7 of the main paper and we find similar takeaways even when using coefficient of variation as a measure of inequality.

## D    ADDITIONAL REBUTTAL RESULTS

We ran the following additional experiments during the rebuttal phase:

- Numbers on x axis for Figure 3 are shown in Figure 13. Figures 4 and 6 compare models with same number of neurons in the final layer and hence trends shown with fraction on the x-axis will be exactly the same with absolute numbers on the x-axis.
- Corresponding diffused redundancy (DR) ablations for Figures 3,4,6. These are shown in Figures 10,11,12 respectively. This should allow easy interpretation of which models/configurations have higher diffused redundancy (lines that are more outside have higher DR). For example Figure 11 clearly shows higher diffused redundancy in models trained on larger upstream datasets (here ImageNet21k) since these curves lie more on the outside of same model's curves for ImageNet1k.
- Performance comparison between randomly chosen neurons and PCA based projection. This is shown in Figure 14. We find that PCA closely follows performance of randomly chosen neurons for CIFAR10 and Oxford-IIIT-Pets. However PCA outperforms randomly chosen neurons on CIFAR100 and Flowers, both of which have more number of classes.
- Results on ImageNet1k and ImageNetV2. We report results on the harder task of ImageNet1k and an even harder task of generalizing to distribution shifts like ImageNetV2 in Figure 15. We find that when randomly dropping neurons, the model is still able to generalize to ImageNet1k with very few neurons, *i.e.*, the phenomena of diffused redundancy

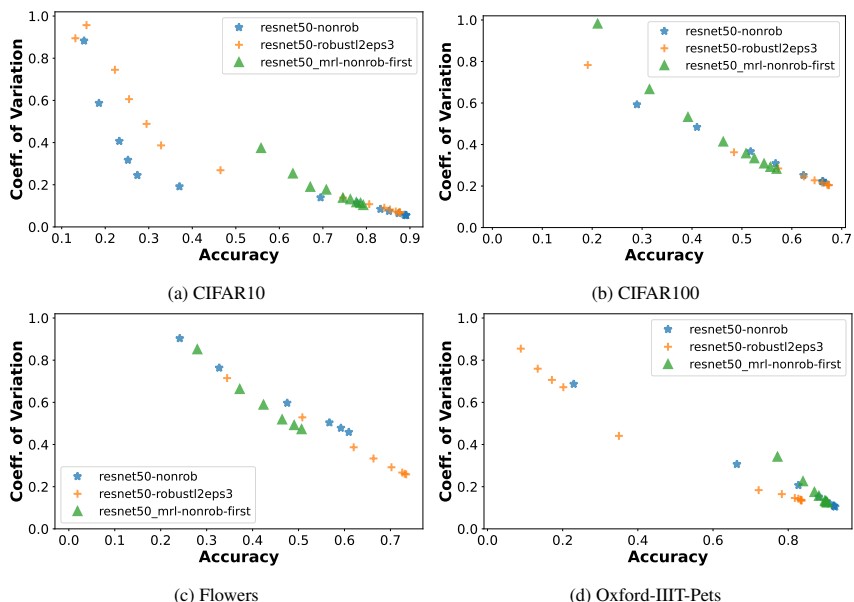

Figure 9: **[Coefficient of Variation As We Drop Neurons]** We see a similar trend as reported in Fig 7 of the main paper where inequality increases as we drop neurons for all models on all datasets.

observed for smaller datasets, also holds for harder datasets. Interestingly we also observe that the accuracy gap between ImageNet1k and ImageNetV2 is maintained even as we drop neurons.

•

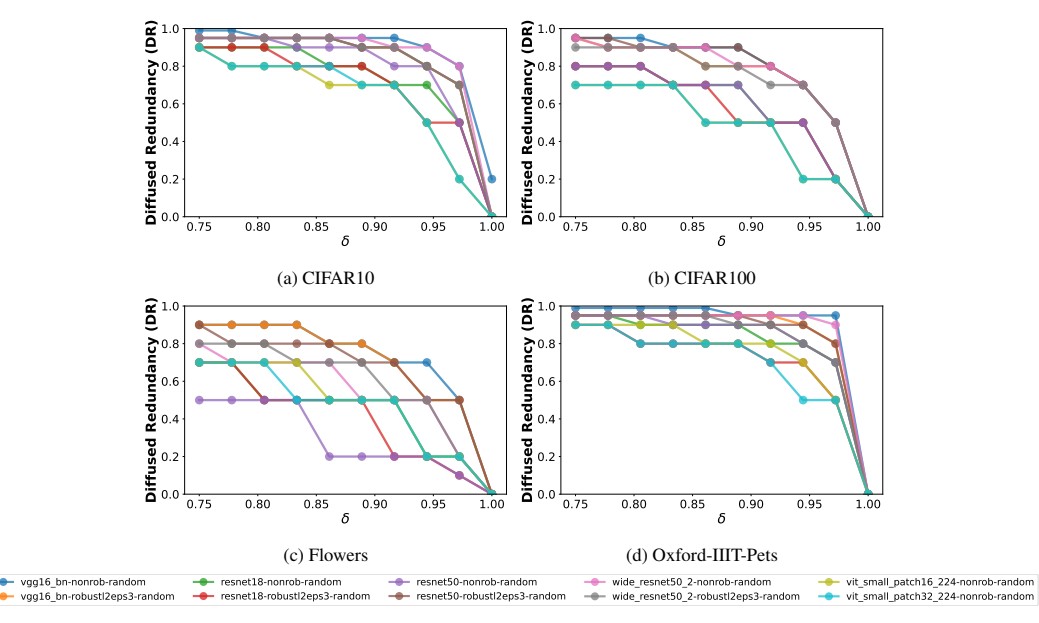

Figure 10: **[Comparisons Across Architectures For Downstream Task Accuracy]** All models shown here are pre-trained on ImageNet1k. This Figure shows corresponding diffused redundancy values for Figure 3 different $\delta$ values. We see that diffused redundancy exists across architectures, and the trend observed in Figure 1c&1a regarding adversarially trained models also holds here as models curves that are more "inside" are the ones trained with standard loss.

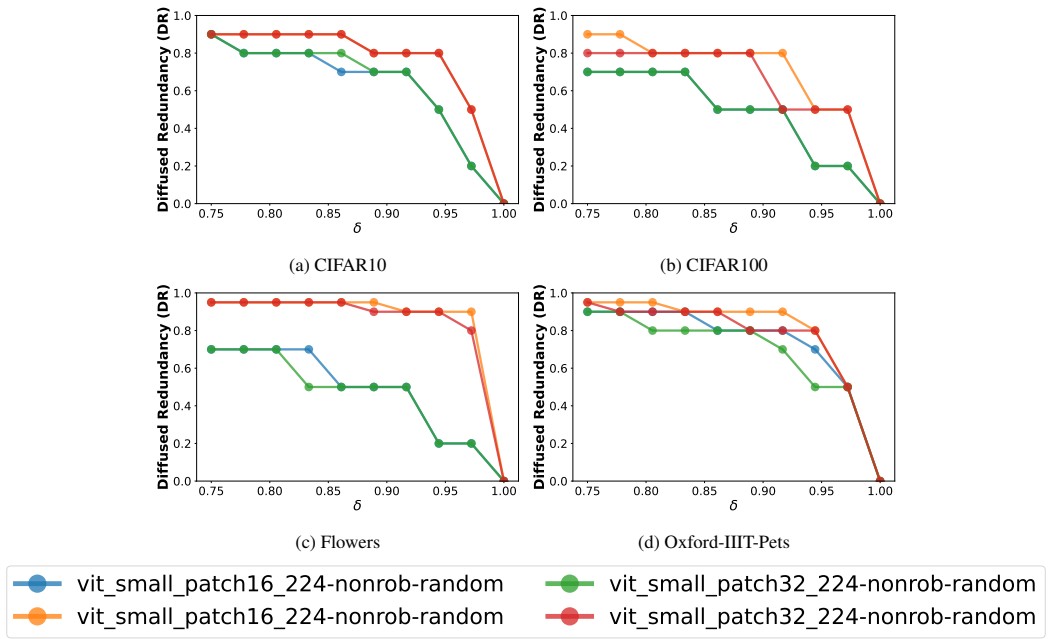

Figure 11: **[Comparison Across Upstream Datasets]** We see that degree of diffused redundancy depends a great deal on the upstream training dataset, in particular models trained on ImageNet21k exhibit a higher degree of diffused redundanacy, although the differences in the degree of diffused redundanacy are downstream task dependent

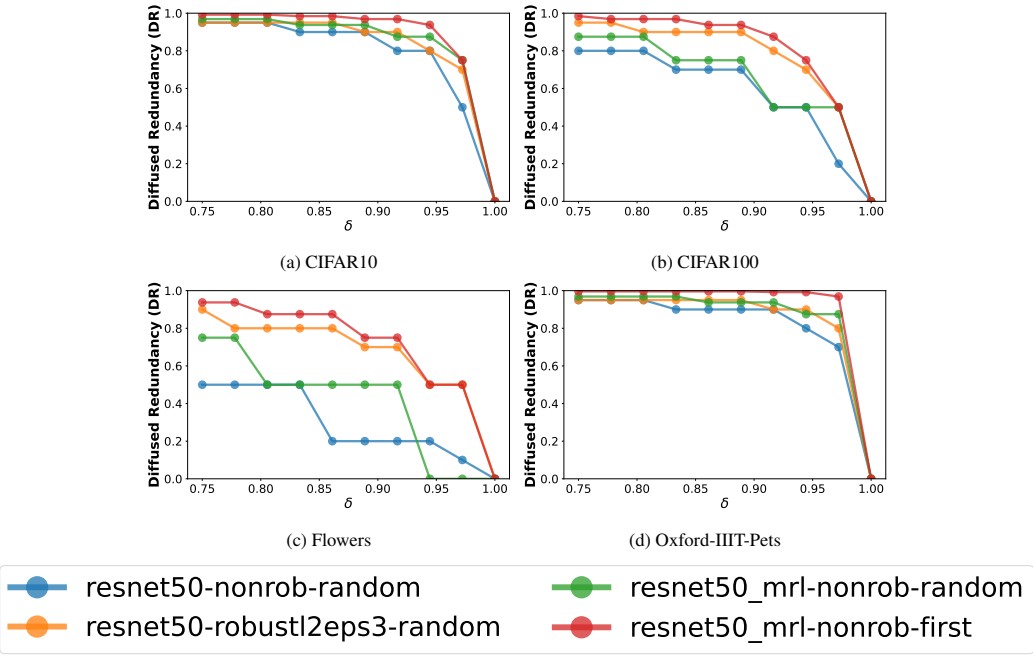

Figure 12: **[Comparison of Diffused Redundancy in MRL vs other losses]** Here we compare ResNet50 trained using multiple losses including MRL (Kusupati et al., 2022). Red line shows results for part of the representation explicitly optimized in MRL, whereas green line shows results for parts that are picked randomly from the same representation. Even the MRL model shows a significant amount of diffused redundancy despite being explicitly trained to instead have structured redundancy. This figure shows diffused redundancy (DR) for all plots in Figure 6.

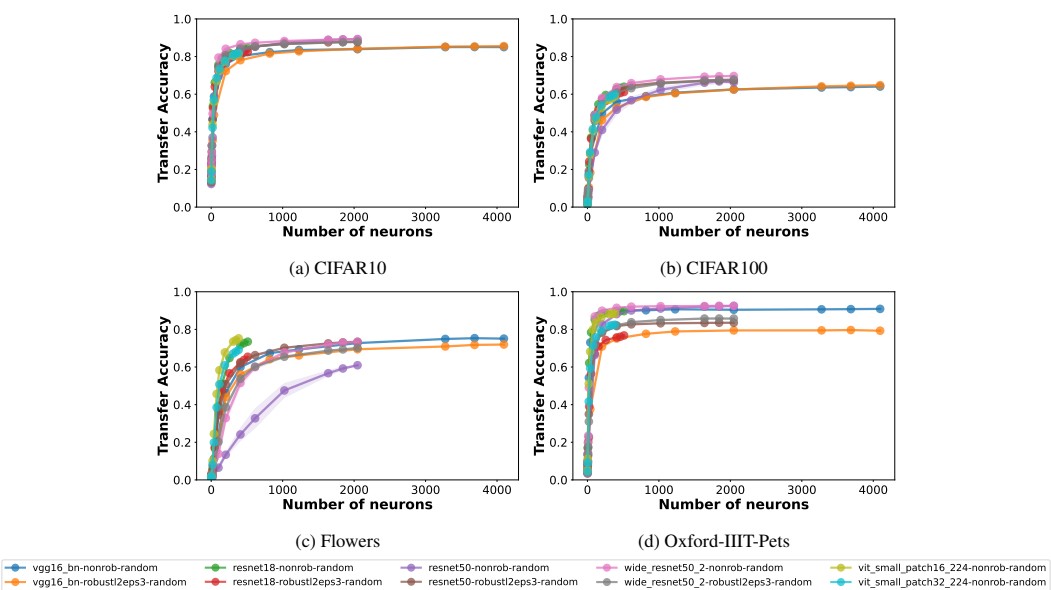

Figure 13: **[Comparisons Across Architectures For Downstream Task Accuracy]** This shows the same plots as Figure 3, except showing absolute number of neurons on the x-axis

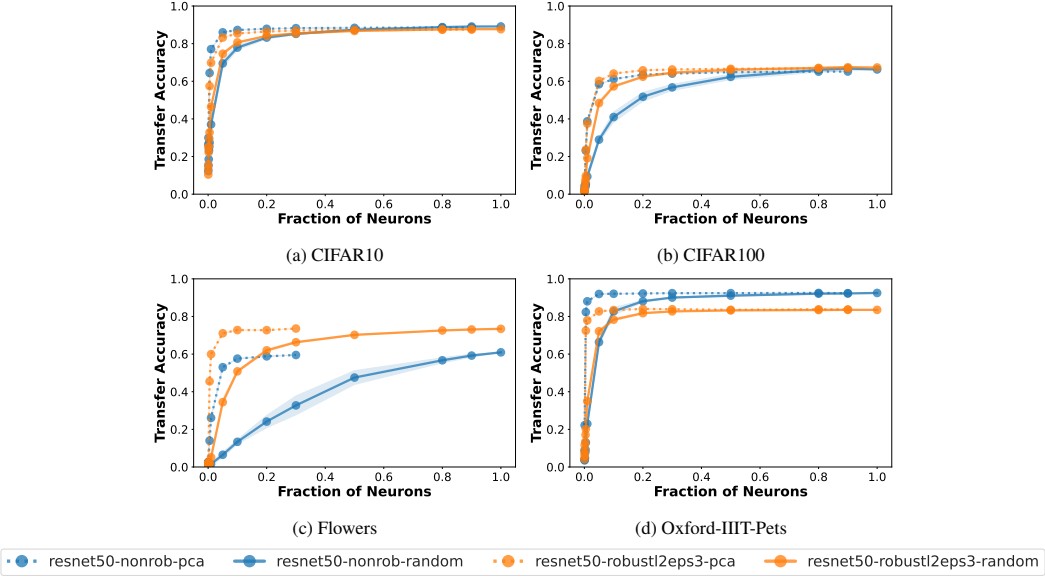

Figure 14: **[PCA vs Diffused Redundancy Comparison]** We compare PCA-based dimensionality reduction to randomly choosing subsets of neurons on ImageNet pretrained ResNet50 (both nonrob and one trained with adversarial training). We find that for some downstream tasks PCA performance closely follows performance obtained via random neurons. For tasks with large number of classes (CIFAR100 and Oxford-IIIT-Pets), PCA outperforms randomly chosen neurons. We also interestingly see that the gap between PCA and random is significantly lesser for the adversarially trained model.

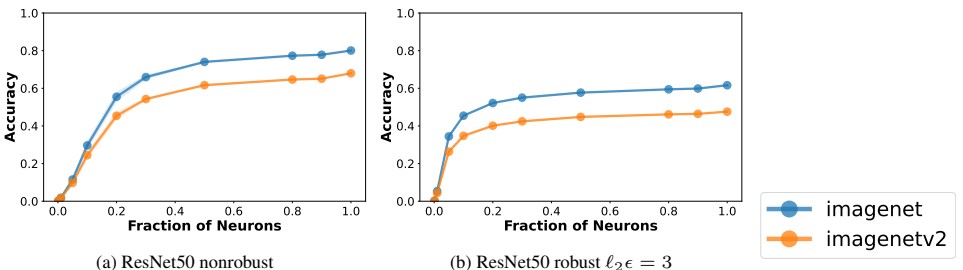

(a) ResNet50 nonrobust

(b) ResNet50 robust $\ell_2\epsilon = 3$

Figure 15: **[Performance on ImageNet1k and ImageNetV2]** We check for performance of randomly chosen subsets of neurons on harder tasks like ImageNet1k and also check for performance differences on a distribution shift, *i.e.* ImageNetV2 Recht et al. (2019). We find that diffused redundancy holds for both these tasks. Additionally we see that randomly dropping neurons still preserves the accuracy gap between ImageNe1k and ImageNetV2.

