# OpenReview forum: "Learned Neural Network Representations are Spread Diffusely with Redundancy"
_ICLR.cc/2023/Conference — Submitted to ICLR 2023_

### Official Review · Reviewer_tTBB · 2022-10-24

**Confidence:** 3
**Correctness:** 3
**Technical Novelty And Significance:** 3
**Empirical Novelty And Significance:** 2
**Recommendation:** 6

**Clarity, Quality, Novelty And Reproducibility:**

Overall, the paper is clear. The point that the authors want to make (that it is possible to drop a large fraction of neurons from the network and still retain downstream performance) is clear, and the paper overall is easy to read. As a minor comment, I would suggest the authors rearrange the figures in page 6, so that they don't cut in the text.

Regarding originality, as stated above the authors make an interesting observation, which however is not that surprising in my opinion. I believe that further analysis on the implications of this observation is needed to fully support this work.

Finally, the authors provide in the Appendix the necessary hyperparameters to reproduce their results.

**Strength And Weaknesses:**

Strengths:

- The paper poses an interesting question overall, in that the authors consider whether the learned representations have any particular structure in how information is redundant in them, and demonstrate that the representations produced by the network are significantly redundant.

- The experimental portion of the paper demonstrates the claim that the authors make, regarding redundancy in the representations of the network. More specifically, it can be seen that in most cases, the downstream accuracy in a given task decays slowly, as the number of neurons in the representation is decreased. The authors also perform a large set of experiments to show what precisely influences diffused redundancy (varying upstream/downstream tasks, layer width and existence of adversarial training).

Weaknesses:

- While I agree with the core observation of the paper, I also don’t find it particularly surprising. It is expected that a large part of the representation produced by the network is redundant, and can be removed without significantly harming the performance of the network. The main point the authors make is that this can be done by removing random neurons of the representation, which shows that this redundancy is spread across the entire representation. However, one can reasonably expect that, since the representation is still in high dimensions, the samples will still be well separated, with only a few neurons kept. To improve this, I believe that the authors should further elaborate on the implications of their observation in the introductory parts of the paper.

- In the experimental section, a simple baseline one can compare against is keeping only the principal directions of the data, via PCA. While this is more complex than simply dropping neurons (since it requires a linear combination of the weights) I believe it would be useful to include this comparison as well.

- When comparing downstream tasks, I think it would also be useful to include the evaluation that arises when the downstream task is the same as the upstream (for example, ImageNet-1K). I think this would be useful to demonstrate whether the observation still holds in a more complicated downstream task.

- The authors include a section in their paper that touches upon issues of fairness that may arise by minimizing the size of the representations of the model. However, I think this part of the paper should be further extended. As it is, it seems to be somewhat disjoint from the rest of the paper. This can easily be an important topic for future work, so I think it should be highlighted further.


**Summary Of The Paper:**

The authors of this paper examine the notion of representation redundancy in large networks, and more specifically how it is diffused across the neurons of the representation. The authors quantify the measure of representation redundancy they use and demonstrate that dropping a large amount of neurons from the last layer randomly only affects the downstream accuracy by a small amount. The authors evaluate their observation on pretrained models on ImageNet, and demonstrate the level of redundancy present in these pretrained representations.

**Summary Of The Review:**

Overall, I think this paper makes an interesting observation, which however requires further analysis on its implications. Currently, I think that there needs to be a better understanding on how the observation made by the authors can be used. Elaborating on the points I mentioned above would help improve this paper.

Update post rebuttal: I have raised my score after the authors' response.

---

> ### Author Response · Authors · 2022-11-18
> **Response to reviewer tTBB: Thank you for your feedback! (1/n)**
>
> We thank the reviewer for their time and thoughtful feedback. We address their concerns below:
>
> ### **Implications of randomly removing neurons**
>
> We agree that having *a possible projection* of the existing layer onto a smaller dimension that performs close to the whole layer would not have been a surprising result. However, we would like to emphasize that picking a *totally random* set of neurons and still having similar performance as the full set of neurons is quite surprising. Our finding implies that the lower dimensional projection formed by *any* subset of randomly selected neurons captures similar amounts of variance captured by the whole layer (hence the name, diffused redundancy). When we conducted the comparison with PCA as suggested by you (see below) we saw that for some downstream tasks, accuracy obtained by training a linear classifier on the projection of the representation on the top k principal components performs very similarly to randomly picking k neurons (our work). This suggests that for these datasets, randomly sampled k neurons capture roughly the same amount of variation in the data as the first k principal components.
>
> Additionally, our results imply that efforts aimed at studying individual neurons are ultimately futile since one can perform similarly efficient computations even by randomly throwing away certain neurons.
>
> ### **Baseline comparison with PCA**
>
> Thank you for this suggestion! We have added a comparison with projections along the first $k$ PCA dimensions (where we vary $k$ the same way we varied random subsampling of neurons in our experiments). We find that PCA-based projection closely follows the performance obtained by random pruning of neurons (ours) for datasets with lesser classes (CIFAR10 and Pets) while PCA outperforms randomly sampled neurons for datasets with large number of classes (CIFAR100, Flowers). We also see that the difference between the performance of PCA and random neurons is much lesser for the adversarially trained model. These results are shown in Figure 14 in Append D. We believe these are very interesting results since they imply that for CIFAR10 and Pets, random subsets of neurons capture similar amounts of variations in the data as the principal components. While PCA needs to be calculated on the training set of every downstream dataset, random pruning offers a faster, simpler solution.
>
>
> ### **Extensions of Fairness Analysis**
>
> Thank you for appreciating the importance of this analysis! Our goal here was to draw caution to fairness consequences of using pruned representations (both structured ones as in MRL and random ones as in our work). Use of such compact representations can be lucrative since they offer quick downstream training, however, researchers and practitioners should be aware of possible pitfalls of this approach. If you have any suggestions for extending this analysis, we would love to hear them!
>
> We're wrapping up experiments on harder datasets like ImageNet1k and Places365 and will report back with new results very soon.

---

> > ### Comment · Reviewer_tTBB · 2022-11-19
> > **Thank you for your responses.**
> >
> > Thank you for your detailed responses to my concerns.
> >
> > I believe the experiment with PCA is a good addition to demonstrate the core idea of this work, by showing that choosing neurons randomly doesn't perform much worse than choosing them via a projection based approach. The point I wanted to make regarding high dimensions is that it seems reasonable to expect that any projection would work well enough (with high enough probability), not just that one such projection exists. Nonetheless, I can see that with the inclusion of comparisons with PCA this concern is alleviated.
> >
> > I also appreciate the inclusion of Figure 15, with ImageNet experiments which demonstrate that the observation still holds. I would be grateful if the authors could provide results on Places365 per their response to Reviewer 9A8r.
> >
> > Finally, regarding the fairness part, I believe that the main way that this section can be improved is via a more detailed presentation of the fairness literature, along with the metrics used to quantify it (since this is the only part where this literature is mentioned in the paper). I believe that this will make the shift from one topic to the other easier to follow.
> >
> > Overall, my concerns have largely been adressed, so I have raised my score somewhat.

---

> ### Author Response · Authors · 2022-11-18
> **Response to reviewer tTBB: Thank you for your feedback! (2/2)**
>
> ### **Results on harder datasets like ImageNet1k**
>
> That’s a great suggestion! We have added these results for ResNet50 in Appendix D, Figure 15 and find that diffused redundancy still holds for harder tasks. Additionally, based on reviewer 9A8r’s suggestion we compared the performance of randomly chosen subsets of neurons on ImageNet1k’s test set with that of ImageNetV2’s test set and find that accuracy difference between these two datasets is preserved as we drop neurons.
>
> We hope we have answered all your concerns!

---

### Official Review · Reviewer_JaiD · 2022-10-25

**Confidence:** 3
**Correctness:** 3
**Technical Novelty And Significance:** 2
**Empirical Novelty And Significance:** 3
**Recommendation:** 6

**Clarity, Quality, Novelty And Reproducibility:**

# Clarity / Quality
- In Figure 1, it is not stated whether diffused redundancy is using CKA or downstream accuracy as the task. The acronym for adversarial training (AT) is never explicitly defined.
- Needs some proofreading. For example: "hence right more point" was probably supposed to be "hence, the rightmost point".

# Novelty
- I am not familiar enough with the related works to evaluate the novelty.

# Reproducibility
- Code and implementation details are provided.

**Strength And Weaknesses:**

# Strengths
- Demonstrates an interesting property of image classifiers.
- Comparisons are done between many different models.
- Potential drawback of creating a class imbalance is highlighted, although this somewhat contradicts the diffused redundancy hypothesis.

# Weaknesses
- The application to model optimization is limited since this technique only reduces the size of the final layer. There could be some comparison in memory usage, floating point operations, or model size to motivate DR as an optimization technique.
- It is not clear if there is any use for the definition of diffused redundancy (DR) (equation 1), while the definition states that it will be used to rigorously test the DR hypothesis. Figure 1a (downstream accuracy) and 1b (diffused redundancy) appear to convey the same information. The rest of the paper does not use this definition.
- Section 2.2 claims to explain why any random subset will work for downstream tasks. Figure 2 shows that two randomly chosen subsets have high representational similarity to each other. This further confirms that any random subset works, but does it explain why? After seeing that a random subset has a similarity to the full layer in figure 1, it is not surprising that random subsets are also similar to each other.
- In figures 3, 4, and 6 I believe the full representations all have different sizes, but they are not displayed. I am wondering if it is more important to show the absolute number of neurons on the x-axis instead of the fraction.

**Summary Of The Paper:**

# Intro / Section 2
- This paper introduces the property of diffused redundancy: any randomly chosen subset of a layer can perform similarly to the full layer on downstream tasks.
- The degree of diffuse redundancy is investigated for the final layers of image classification models trained on imagenet.
- Two downstream tasks are used to measure diffused redundancy, 1) representational similarity (CKA) between the whole layer and random subsets of the layer, and 2) classification accuracy when using the whole or a random subset of the layer to train a linear classifier for a downstream dataset.
- Redundancy is when a high performance can be accomplished with a small fraction of neurons.
- Diffusivity is when the variability of performance is low for different random subsets of neurons.

# Section 2.1 / Figure 1
- Representational similarity and downstream accuracy are plotted for different sizes and random subsets of the full layer. A comparison is done between four downstream datasets (cifar10/100, flowers, oxford-iit-pets), and two models (ResNet50 with and without adversarial training).
- Degree of diffusion and redundancy depends on the downstream dataset.
- The adversarially trained model demonstrates greater diffused redundancy.

# Section 2.2 / Figure 2
- Plots of representation similarity between randomly chosen subsets.

# Section 3 / Figure 3
- Downstream accuracy vs subset sizes is compared for a larger collection imagenet classification models.

# Section 3.1 / Figure 4
- Downstream accuracy vs subset size is compared for models trained on imagenet1k and imagenet21k.
- The imagenet21k models demonstrate greater downstream accuracy.

# Section 3.2 / Figure 5
- Downstream accuracy vs subset size is compared for models with varying final layer sizes.
- The size of the final layer is proportional to downstream accuracy.

# Section 3.3 / Figure 6
- Downstream accuracy vs subset size is compared for models that are trained with MRL (smaller parts of the representation have structure).
- The MRL-optimized representation performs well for very low numbers of neurons, but is outperformed for larger subsets.

# Section 4 / Figure 7
- The loss in downstream accuracy when reducing the size of the random subset is primarily in a few classes instead of being uniformly distributed.
-This suggests that exploiting diffused redundancy has implications for fairness.

**Summary Of The Review:**

Diffused redundancy is an interesting property of the final layer of image classifiers, but its application is not clear.

---

> ### Author Response · Authors · 2022-11-17
> **Response to reviewer JaiD: Thank you for your feedback! (1/2)**
>
> We thank the reviewer for their time and constructive feedback! We address their comments below:
>
> ### **Application to model optimization**
>
> We would like to emphasize that our goal is *not* model optimization. There has been long-standing research on structured pruning (He et al., 2017; Li et al., 2016) that achieve optimizations (in terms of FLOPS, memory, inference times vs drop in performance) which are way better than randomly dropping features (our work). Our goal, however, is to point to an intriguing property of learned features in commonly used pre-trained DNNs. We would like to emphasize that one can still use diffused redundancy in conjunction with structured pruning methods (since these methods act on initial convolution layers) to get a model that is compact overall. However, this analysis is outside the scope of this paper and is a very interesting direction for future work!
>
> (Li et al., 2016) Pruning filters for efficient convnets
>
> (He et al., 2017) Channel pruning for accelerating very deep neural networks
>
> ### **Definition of Diffused Redundancy**
>
> Fig 1b can indeed be derived from 1a – this is because the definition of diffused redundancy relies on downstream accuracy. We do believe it’s important to precisely state what we’re measuring, hence we wrote down Eq 1 to ensure there’s no ambiguity. However, we do agree that in this case Eq 1 can appear to be more convoluted and it’s perhaps easier for the reader to understand what we mean if we state it in plain English. If you think it aids readability, we’re happy to move Eq 1 to the appendix and replace it with the following definition:
>
> *For a given task T, diffused redundancy is defined as the fraction of neurons that can be randomly discarded to obtain a performance level within a $\delta$ fraction of the performance of the entire layer*
>
> ### **Not surprising to see high similarity between small subsets when we see high similarity between subsets and the whole layer**
>
> Thanks for bringing this up! This raises a fundamental question about the transitivity of CKA as a similarity measure. That is, if $\text{CKA}(X,Z)$ is high and $\text{CKA}(Y,Z)$ is high, does it imply $\text{CKA}(X,Y)$ is also high? To us, it’s not immediately obvious that this is necessarily true since CKA is not supposed to be transitive by design. This is because CKA captures the similarity of pairwise similarities (for Linear CKA, which we use in our paper similarity is based on cosine distance) between two representations – thus Linear CKA between $X \in \mathbb{R}^{n \times d1}$ and $Z \in \mathbb{R}^{n \times d2}$ looks something like:
>
> $\text{CKA}(X,Z) = \frac{\text{Tr}(XX^{T}ZZ^{T})}{C_{XZ}}$
>
> Similarly CKA between $Y \in \mathbb{R}^{n \times d3}$ and $Z \in \mathbb{R}^{n \times d2}$ is:
>
> $\text{CKA}(Y,Z) = \frac{\text{Tr}(YY^{T}ZZ^{T})}{C_{YZ}}$
>
> Where we treat $C_{XZ}$ and $C_{YZ}$ as representation-specific constants for the purpose of this analysis (these are normalizations to ensure CKA is invariant to isotropic scaling). To show that high $\text{CKA}(X,Z)$ and high $\text{CKA}(Y,Z)$ does not necessarily imply high $\text{CKA}(X,Y)$ it suffices to show a counterexample:
>
> Say,
> $$\text{CKA}(X,Z) = \frac{\text{Tr}(\begin{pmatrix} 1 & .. & .. & .. \\\ .. & 1 & .. & .. \\\ .. & .. & 0 & .. \\\ .. & .. & .. & 0 \end{pmatrix})}{C_{XZ}} $$
>
> And,
> $$\text{CKA}(Y,Z) = \frac{\text{Tr}(\begin{pmatrix} 0 & .. & .. & .. \\\ .. & 0 & .. & .. \\\ .. & .. & 1 & .. \\\ .. & .. & .. & 1 \end{pmatrix})}{C_{YZ}} $$
>
> Without loss of generality, say, $C_{XZ} = C_{YZ} = C_{XY} = 4$, giving us $\text{CKA}(X,Z) = \text{CKA}(Y,Z) = 0.5$
>
> This means that the first two elements in X have similar neighborhoods as the first two points in Z but other points have very different neighborhoods in X and Z. And the last two points in Y and Z have similar neighborhoods but all other points differ in their neighborhoods. Thus, the points contributing to "high" similarity of X and Z are disjoint from the points contributing to the "high" similarity of Y and Z. Thus when computing similarity between X and Y, we're likely to get:
>
> $$\text{CKA}(X,Y) = \frac{\text{Tr}(\begin{pmatrix} 0 & .. & .. & .. \\\ .. & 0 & .. & .. \\\ .. & .. & 0 & .. \\\ .. & .. & .. & 0 \end{pmatrix})}{C_{YZ}} $$
>
> Thus, we get a case where $\text{CKA}(X,Z) = \text{CKA}(Y,Z) = 0.5$ but $\text{CKA}(X,Y) = 0$.
>
> While this is certainly just one example, it shows that transitivity of CKA is not guaranteed. Hence the goal of Section 2.2 is to empirically verify that individual subsets are also similar to one another.

---

> ### Author Response · Authors · 2022-11-17
> **Response to reviewer JaiD: Thank you for your feedback! (2/2)**
>
> ### **Absolute numbers in Figures 3,4,6**
>
> Figures 4 and 6 compare models with same number of neurons, thus the plots with number of neurons on x-axis will look exactly like the current plots. Figure 3 indeed compares models that have different layer sizes and perhaps number of neurons on x-axis could be more informative here. We also debated this when writing the paper, however, absolute numbers can be misleading when comparing across models since the full layer sizes can be drastically different. For example, comparing same number of neurons from VGG16 (representation of size 4096) and ViTS-16 (representation size of 384) would be hard to interpret since for each point on the x-axis (i.e., same number of neurons) there are more neurons being left out for VGG16 and hence it’s natural to expect worse performance than ViT. Nonetheless, we agree with you that it can add value to the reader and have added a plot which mirrors Figure 3 except has number of neurons on the x-axis. This is shown in Appendix D, Figure 13.
>
> Please let us know if our response addressed all the concerns!

---

### Official Review · Reviewer_9A8r · 2022-10-27

**Confidence:** 5
**Correctness:** 3
**Technical Novelty And Significance:** 3
**Empirical Novelty And Significance:** 2
**Recommendation:** 6

**Clarity, Quality, Novelty And Reproducibility:**

See the above section. The paper is highly reproducible with code in the supplementary.

**Strength And Weaknesses:**

I will go sequentially with the strengths and weaknesses

Strengths:
1) The Paper is well-motivated, decently written, and easy to understand.
2) Systematic experimentation is done to understand redundancy in the representation of 4 smaller-scale datasets using ImageNet pre-trained models
3) Experimental findings are consistent across the downstream datasets, pre-training data, architecture, and learning algorithms (normal training, MRL, adversarial).
4) The trade-off between representation size and fairness is very important (and also a good extension to MRL -- Kusupati et al., NeurIPS 2022)
5) The choice of definition of diffused redundancy with the assumption of $\delta=0.9$ is a good starting point.


Weakness:
1) The existence of redundancy is not novel, in fact, MRL has also discussed it in their appendix implicitly (diffused redundancy experiments with particular feature size) - however, concretizing it is a good first step taken by this paper
2) Baseline with correlation regularizer should’ve been discussed - “Reducing Overfitting in Deep Networks by Decorrelating Representations”
3) The paper should cite and discuss papers with structured pruning which are very relevant to the potential redundancy in the hidden layer.
4) Figure 1 should also compare algorithms, that is CKA v/s fraction to better understand if some algorithms have less redundancy than others.
5) Plain CKA numbers are a bit hard to understand. That is, for certain tasks, a CKA of 0.7 may be good, but for other, it may not be. Therefore, a scatter plot downstream task performance v/s CKA at different neuron fractions perhaps could be more informative.
6) On Figure 4: When we are comparing the pre-training datasets, it is difficult to draw conclusions if the accuracy difference for full set of neurons differs quite a bit (which is the case in this figure)
7) The figures could be made more easily readable, it is hard to go through the densely packed plots.
8) Lastly, while the experiments are a good first step, the datasets often are easy -- I would like to see two sets of ablations a) with the variation of $\delta$ from 0.8 to 0.99 and b) experiments on harder datasets like ImageNetV2 and Places365. This would make the paper extremely strong and would help any future papers built on top of it.

I am very excited about this paper and want to have a conversation with the authors through rebuttal and revision. I want the authors to take the weakness in good spirit to help improve the paper to make it worth a strong publication


**Summary Of The Paper:**

Pretrained deep representations from large datasets have a plethora of applications across vision and natural language. Understanding how information is encoded across the dimensions in the representation is important to design better algorithms for the respective downstream tasks.

This paper discusses that the learned representations exhibit significant redundancy. That is, not all the dimensions are needed for getting a decent performance on the downstream tasks, and propose a notion of diffused redundancy. The paper explores the existence of diffused redundancy across the architectures, upstream and downstream tasks, and learning algorithms. Based on the experiments, the paper concludes that the entire layer is not necessary for the downstream task, however, providing caution on how smaller representations sizes may cause unfair decisions

**Summary Of The Review:**

Good step towards quantifying the redundancy in the representations. Experiments on small scale datasets and some missing baselines. However, a strong effort towards something useful at scale.

---

> ### Author Response · Authors · 2022-11-17
> **Response to reviewer 9A8r: Thank you for your positive comments and constructive feedback! (1/n)**
>
> We thank the reviewer for their thoughtful feedback and for positively engaging with our work! We address some of the concerns below.
> ### **Baseline comparison with DeCov regularizer**
>
> Thank you for pointing us to this very relevant work! We, unfortunately, could not find any publicly available implementation of this paper or any trained model weights with this regularizer. We contacted the authors in case they had trained models and they promptly responded with their code, but unfortunately, they did not have model weights (which seem to have been lost to time). Since this paper was published in 2016 the implementation was in Caffe so we decided to re-implement this on our own in PyTorch. However, it will take us more time than the rebuttal phase to ensure correct implementation of this regularizer for a fair comparison. We will have the results in time for the camera ready and will include this comparison in the final version of the paper.
>
> ### **Relationship to structured pruning**
>
> Thank you for pointing this out. Indeed prior works on structured pruning are very relevant to our work. We already cite (Li et al, 2016) under related work and have added a citation to (He et al., 2017) in the revised version with added discussion on how these works differ from diffused redundancy (text shown in red). A major goal of structured pruning is to increase the efficiency of DNN inference and this is achieved by pruning entire channels/filters of convolution layers. This ensures that there are 1.) lesser matrix computations to be done, and 2.) whatever matrix multiplications remain are still dense and thus can be efficiently computed using libraries optimized for dense operations (eg: CUDA/BLAS). Additionally, these methods are particularly tailored towards CNNs, since they operate only on convolution layers. Our work differs in both the goals and the mechanism of pruning. Firstly, as opposed to a particular application, our primary goal is to better understand the nature of learned features. Secondly, we drop random features, whereas works on structured pruning perform magnitude or feature-selection-based pruning of features. This also allows us to perform analyses on models other than CNNs (eg: ViT) and more broadly on all kinds of layers. We think it would be very interesting to combine diffused redundancy with structured pruning to get models that are compact in both the initial (convolution) layers and final (fully-connected) layers. However, this is outside the scope of this paper and we leave it as an interesting avenue to pursue for future work!
>
> (Li et al., 2016) Pruning filters for efficient convnets
>
> (He et al., 2017) Channel pruning for accelerating very deep neural networks
>
> If you have any more relevant citations, please let us know and we’d be happy to add them to the paper!
>
> ### **CKA numbers are task and performance agnostic**
>
> We would like to emphasize that CKA numbers are task agnostic and should be interpreted independently of downstream performance. Two networks can have close to random performance on a dataset but still have high CKA if they place every data point in similar neighborhoods across the two representations. Thus we use CKA as a separate measure from downstream performance to evaluate diffused redundancy.
>
> ### **Hard to interpret Figure 4 due to full layer accuracy differences**
>
> We totally agree with this concern and thank you for bringing this up. Diffused redundancy as defined in Eq 1 should be interpretable regardless of the absolute value of full layer’s performance. We’ve thus added a plot showing diffused redundancy for both models with varying deltas in Appendix D. We hope this gives a more easily comparable interpretation of Figure 4.
>
> ### **Ablations of Delta**
>
> We do report these ablations in Figure 1b and 1d. We have added ablations for Figures 3,4 and 6 in Appendix D shown in Figures 10, 11 and 12 respectively.

---

> > ### Comment · Reviewer_9A8r · 2022-11-19
> > **Thanks for the Rebuttal - Delayed Response**
> >
> > Thanks to the authors for their rebuttal. I will get back to this in a day or so, sorry for the delay.

---

> > ### Comment · Reviewer_9A8r · 2022-11-22
> > **Response to rebuttal**
> >
> > Dear Authors,
> >
> > Thanks for your rebuttal. Here are my comments:
> >
> > 1) On DeCov: I would highly appreciate it if authors can add this training mechanism in the final version. It is an important baseline, and its absence is the reason to retain the final rating.
> >
> > 2) On CKA: While we can construct a counterexample where two networks can have a very high CKA yet close to random prediction performance, in this case, it can still be insightful to have a scatter plot of downstream accuracy on one axis and CKA on the other. This is because of the fact that when using the full layer, the model doesn’t result in close to random accuracy.
> >
> > Also, when we have no idea how to read a metric, grounding it and saying what it stands for helps in a quantitative sense -- I don't know how significant 1% is for a sentiment classification task but know how valuable it is for Imagenet - given my background -- it helps people like me to understand more.
> >
> > 3) Figure 11 seems to have a typo in the legend. I believe the color scheme is the same as that used in the main paper (Fig4), but it needs to be corrected.

---

> ### Author Response · Authors · 2022-11-18
> **Response to reviewer 9A8r: Thank you for your positive comments and constructive feedback! (2/2)**
>
> ### **Results on harder datasets like ImageNetV2 and Places365**
>
> Thank you for these suggestions! We believe these are two different but equally interesting analyses:
>
> 1. Does the phenomena of diffused redundancy still hold for harder datasets like ImageNet and Places365?
>
> 2. How do models with lesser neurons perform on distribution shifts such as ImageNetV2?
>
> In order to answer these questions we trained linear probes on different subsets of neurons from the final layers of ImageNet on the usual ImageNet1k training set and evaluated on the ImageNet1k eval set *and* on ImageNetV2 (note that ImageNetV2 is only an evaluation dataset and we cannot use it for training, as suggested by the original ImageNetV2 paper (Recht et al.,2019)). We have added these results in Appendix D, Figure 15. We find that when randomly dropping neurons, the model is still able to generalize to ImageNet1k with very few neurons, i.e., the phenomena of diffused redundancy observed for smaller datasets, also holds for harder datasets. Interestingly we also observe that the accuracy gap between ImageNet1k and ImageNetV2 is maintained even as we drop neurons.
>
> Places365 is unfortunately not publicly available anymore (http://places2.csail.mit.edu/download.html) so we contacted the author for a copy and they were kind enough to provide it to us. It will take us a bit more time to finish experiments on Places365 and we will report back with the results as soon as we have them.
>
> (Recht et al.,2019) Do ImageNet Classifiers Generalize to ImageNet?
>
> We hope we have addressed the concerns! Please let us know if you’d like any further clarifications!

---

> > ### Comment · Reviewer_9A8r · 2022-11-22
> > **Thanks**
> >
> > V2 numbers look solid. I would also like to see Places in the final version.
> >
> > I will discuss this with other reviewers but I am leaning toward accepting the paper.
> >
> > Thanks.

---

### Comment · Area_Chair_GnwB · 2022-12-09
**Dropout?**

It seems like at least some of the networks you explore use dropout (VGG-16 https://arxiv.org/pdf/1409.1556.pdf and possibly the ViT models https://arxiv.org/pdf/2010.11929.pdf).  Doesn't dropout essentially force diffuse and redundant representations?  How can the results be interpreted in light of that?

---

### Decision · Program_Chairs · 2023-01-20

**Decision:**

Reject

**Justification For Why Not Higher Score:**

There are several outstanding issues that require additional experiments which the authors were unable to complete during the rebuttal (one requires a significant amount of coding).  This paper is close, but not quite ready for publication.

**Justification For Why Not Lower Score:**

n/a

**Metareview: Summary, Strengths And Weaknesses:**

This paper explores the representations learned by CNNs, and shows that subsets of the neurons in a particular layer can be used to summarize that layer's contributions to the final prediction.  Thus there is a lot of redundancy in the representations.

Reviewers agreed that the experiments were extensive, interesting and the paper was well laid out.  The reviewers raised some important points about the evaluation, specifically suggesting a new baseline (which would be a fair amount of work to implement) and other datasets (one of which has been incorporated).  There was some missing related work, but this seems to have been addressed.

My take is that this is a strong paper, but needs a bit more work before it's ready for publication